# Bioactivity Screening and Gene-Trait Matching across Marine Sponge-Associated Bacteria

**DOI:** 10.3390/md19020075

**Published:** 2021-01-30

**Authors:** Asimenia Gavriilidou, Thomas Andrew Mackenzie, Pilar Sánchez, José Ruben Tormo, Colin Ingham, Hauke Smidt, Detmer Sipkema

**Affiliations:** 1Laboratory of Microbiology, Wageningen University and Research, 6708 WE Wageningen, The Netherlands; hauke.smidt@wur.nl (H.S.); detmer.sipkema@wur.nl (D.S.); 2Fundación MEDINA, Centro de Excelencia en Investigación de Medicamentos Innovadores en Andalucía, Avda. del Conocimiento 34, 18016 Granada, Spain; thomas.mackenzie@medinaandalucia.es (T.A.M.); pilar.sanchez@medinaandalucia.es (P.S.); ruben.tormo@medinaandalucia.es (J.R.T.); 3Hoekmine BV, 3515 GJ Utrecht, The Netherlands; colinutrecht@gmail.com

**Keywords:** sponge-associated bacteria, antibacterial, anticancer, biosynthetic gene clusters, gene-trait matching

## Abstract

Marine sponges harbor diverse microbial communities that represent a significant source of natural products. In the present study, extracts of 21 sponge-associated bacteria were screened for their antimicrobial and anticancer activity, and their genomes were mined for secondary metabolite biosynthetic gene clusters (BGCs). Phylogenetic analysis assigned the strains to four major phyla in the sponge microbiome, namely Proteobacteria, Actinobacteria, Bacteroidetes, and Firmicutes. Bioassays identified one extract with anti-methicillin-resistant *Staphylococcus aureus* (MRSA) activity, and more than 70% of the total extracts had a moderate to high cytotoxicity. The most active extracts were derived from the Proteobacteria and Actinobacteria, prominent for producing bioactive substances. The strong bioactivity potential of the aforementioned strains was also evident in the abundance of BGCs, which encoded mainly beta-lactones, bacteriocins, non-ribosomal peptide synthetases (NRPS), terpenes, and siderophores. Gene-trait matching was performed for the most active strains, aiming at linking their biosynthetic potential with the experimental results. Genetic associations were established for the anti-MRSA and cytotoxic phenotypes based on the similarity of the detected BGCs with BGCs encoding natural products with known bioactivity. Overall, our study highlights the significance of combining in vitro and in silico approaches in the search of novel natural products of pharmaceutical interest.

## 1. Introduction

The fight against cancer and infectious diseases are two of the main challenges the scientific and medical community have been facing during the past decades. A large proportion of compounds for the treatment of human ailments throughout history, from traditional remedies to current-day pharmaceuticals, belonged to natural products [1]. However, the decline in the discovery rates of new bioactive molecules led to skepticism on the potential of drug development from natural products [2]. Nowadays, in the light of the urgent need for new drugs, particularly those with anticancer and anti-infective properties, increasing attention is returning towards natural products and more specifically, from marine sources [3,4]. The field of marine drug discovery has been growing over the past 20 years, with currently almost 35,000 research articles on natural products of marine origin [5]. This is also highlighted by the number of novel secondary metabolites being elucidated per year (1554 in 2018) and a significant number of marine-derived drug candidates under clinical trials or pending approval [6,7,8].

Despite the intensive research effort, the marine environment can be considered rather underexplored for prospecting bioactive molecules in comparison with terrestrial ecosystems [9,10]. It encompasses an enormous biodiversity and chemical richness covering approximately 70% of the Earth’s surface [11]. From the marine biosphere, sponges are standing out, often termed as “chemical factories” or “gold mines” owing to the production of metabolites with various pharmaceutically interesting biological activities [12,13,14]. The ecological role of these molecules is linked to the adaptation to the marine ecosystem where chemical defense is a survival mode for sponges against predators, competitors, fouling organisms, and infectious microbes [15,16].

An integral part of marine sponges is their symbiotic microorganisms, which comprise up to 40% of their total volume, forming close associations with the host [17,18], a concept recently defined as “sponge holobiont” [19]. Microbial associates have been found to contribute significantly to many functions of the host, including nutrition, health, and defense [17,19,20,21]. In fact, there is a growing amount of experimental evidence that many bioactive substances produced by sponges are of bacterial origin, suggesting sponge-associated bacteria as the real producers rather than the host itself [12,22,23,24,25]. Two main findings support this theory, the structural similarity of molecules derived from taxonomically distant taxa and the case of bioactive compounds found in sponges, but known to be synthesized by bacteria, such as polyketides and non-ribosomal peptides [20,26,27,28,29,30].

More than 60 microbial phyla inhabit marine sponges [31], from which the most predominant ones are the Proteobacteria, Chloroflexi, Thaumarchaeota, Cyanobacteria, Candidatus Poribacteria, Acidobacteria, and Actinobacteria [21,31]. Symbiont-derived compounds have displayed a wide spectrum of biological activities including antimicrobial, anticancer/cytotoxic, anti-inflammatory, antifouling. The majority of sponge-associated microorganisms have been found to synthesize compounds with antimicrobial, antitumor and cytotoxic properties [32,33]. In the sponge holobiont, the most prominent bacterial producers of antimicrobials are members of the Actinobacteria phylum, followed by Proteobacteria, Firmicutes, and Cyanobacteria [27,32,34,35]. The most prolific sources of antimicrobial compounds among bacteria isolated from sponges are the following genera: *Streptomyces*, *Pseudovibrio,* and *Bacillus* [34]. Recently, the study of a novel symbiotic bacterium *Bacillus tequilensis* of the marine sponge *Callyspongia diffusa* led to the discovery of an antibiotic agent active against multidrug resistant *Staphylococcus aureus* [36]. Another recent example was the elucidation of three new flavonoids from sponge-derived *Streptomyces* sp. with antimicrobial activity [37]. Likewise, most anticancer (i.e., cancer-preventive, antitumor, cytotoxic) molecules produced by sponge-bacteria associations are derived from Proteobacteria (γ-Proteobacteria), Actinobacteria, and Firmicutes [32]. Some recent examples of natural products isolated from sponge-dwelling bacteria with antitumor and/or cytotoxic potential are nocardiotide A produced by *Nocardiopsis* sp. associated with *Callyspongia* sp. [38] and the dipeptide cyclo(-Pro-Tyr) produced by *Bacillus pumilus* associated with *Callyspongia fistularis* [39].

Gaining full access to the actual producers via cultivation and discovering new lead structures are two of the main bottlenecks that exist in the drug discovery pipeline [1,40]. The biosynthetic capacity of even the most well-characterized microbes has been underexplored, since a vast number of biosynthetic gene clusters (BGCs) remain cryptic [40,41,42]. A promising strategy to expand our understanding of the true metabolic potential of microorganisms and increase the probability of finding new molecules involves in silico genome mining of cultivable isolates [42,43]. In the current study, our aim was to (1) in vitro assess the activity of sponge-associated bacteria against various human pathogens and cancer cells; (2) examine their secondary metabolite biosynthesis potential via genome mining; and (3) identify the BGCs potentially involved in the observed bioactivity by gene-trait matching.

## 2. Results

### 2.1. Genome Characteristics

The genomes of 21 strains isolated from six different sponge species (*Aplysina aerophoba*, *Acanthella acuta*, *Corticium candelabrum*, *Chondrilla nucula*, *Ircinia* sp., and *Petrosia ficiformis*) in previous studies [44,45] were analyzed here. Eight genomes (Aa3_DN64_1D3, Aa3_Str.68_7G12, Acac_Ps_AB113, Cn_Ps_AB111, Irc_Ps_AB108, Pf1_DN206_4B7, Pf1_Ps_8H04_1, Pf1_Ps_8H06) were generated by Versluis et al. [44] and the rest were produced in this study (Table 1). The average number of contigs was 46 and the average coverage per base of the draft assemblies was 219×. The genome size ranged from 2.6 to 7.2 Mbp with a GC content between 32.9 and 72.9% and the average number of genes being 4654. Genome completeness was 98.7% on average and contamination was less than 1.5% in all cases. More detailed information on the genome assembly metrics can be found in Appendix A.

### 2.2. Strain Identification and Phylogeny

All strains analyzed in this study were identified based on both 16S ribosomal RNA (rRNA) gene sequences and single-copy marker gene analysis. The 16S rRNA gene-based taxonomic assignment of the sponge-associated strains according to NCBI database [46] was confirmed by the Genome Taxonomy Database (GTDB) (Table 1 and Appendix A) [47,48]. Four main clades were formed representing the following phyla: Proteobacteria, Actinobacteria, Bacteroidetes, and Firmicutes (Figure 1). Out of the 21 strains, 16 strains belonged to Proteobacteria, three strains were classified as Actinobacteria, and one strain each represented Bacteroidetes and Firmicutes, respectively. Within the Proteobacteria, there were strains most closely related to *Pseudovibrio*, *Bradyrhizobium*, *Ruegeria*, *Microbulbifer*, *Acinetobacter,* and *Psychrobacter* strains. In the case of Actinobacteria, members of the genera *Rhodococcus*, *Brevibacterium,* and *Janibacter* were identified. Moreover, two strains were affiliated with *Aquimarina* and *Bacillus* genera, respectively (Figure 1).

### 2.3. Antimicrobial Activity Screening

Crude extracts of axenic bacterial cultures were tested for their antibacterial activity against a panel of Gram-positive and Gram-negative bacteria (*E. coli*, *K. pneumoniae,* and *S. aureus* MRSA). Only one strain (Aa3_DN216_4B10_1) showed significant growth inhibition (61%) of *S. aureus* MRSA. According to both 16S rRNA gene- and whole genome-based taxonomy, the strain Aa3_DN216_4B10_1 was closely related to *Rhodococcus erythropolis*, member of the Actinobacteria (Figure 1). The antifungal activity of the extracts was assessed against *C. albicans* and *A. fumigatus*. Growth inhibition of the fungi was not observed for any of the extracts under the tested conditions (data not shown).

### 2.4. Anticancer Activity Screening

The cytotoxicity of the bacterial crude extracts was determined on human skin (A2058), lung (A549), liver (HepG2), breast (MCF7), and pancreas (MiaPaca2) cancer cells. In general, the majority of extracts (76.2%) were mostly effective against HepG2 cells exhibiting moderate to high cytotoxicity (Figure 2). In addition, almost 20% of the extracts resulted in cell death of more than 50% of the A2058, A549, and MiaPaca2 cancer cells, while very weak activity was observed against the MCF7 cell line. Four extracts (Pf1_DN64_8G1, Aa3_DN64_1D3, Aa3_DN73_5E10_2, and Aa3_DN213_3F7) had the highest activity, causing more than 50% of cell death of at least two of the cancer cell lines (Figure 2). These were obtained from isolates belonging to Proteobacteria and Actinobacteria and more specifically to the genera *Pseudovibrio, Psychrobacter,* and *Brevibacterium* (Figure 1). No correlation was observed between phylogenetic proximity of tested strains and their levels of cytotoxicity, except for Aa3_DN64_1D3 and Pf1_DN64_8G1, which both belonged to *Pseudovibrio* and were highly active (Figure 1 and Figure 2).

### 2.5. Biosynthetic Gene Cluster Profiling

Genomes of the isolates were mined for BGCs, and a total of 153 BGCs belonging to 28 different BGC types as classified by antiSMASH (antibiotics and Secondary Metabolite Analysis Shell) [49], were identified. In terms of absolute abundance, the majority of identified BGCs was predicted to encode bacteriocins (*n* = 24), non-ribosomal peptide synthetases (NRPS) (*n* = 19), beta-lactones (*n* = 18), terpenes (*n* = 18) and siderophores (*n* = 10). Similarly, the same categories displayed the highest frequency being detected in almost 50% of the genomes (Figure 3). Considering both the genome size and the abundance of BGCs (Appendix A), the strains with the highest secondary metabolite biosynthesis potential were the following: Aa3_DN216_4B10_1, Aa3_DN213_3F7, Aa3_DN64_1D3, Pf1_DN64_8G1 and, Aa3_DN30_1H2, all from the Actinobacteria, Proteobacteria, or Firmicutes.

### 2.6. Gene-Trait Matching

In silico prediction of the secondary metabolite BGCs was combined with the experimental data obtained from the bioactivity screening assays for gene-trait matching (GTM) aiming at the correlation of genetic features with specific phenotypes. Strains that showed both high secondary metabolite biosynthetic potential and in vitro bioactivity were selected for further analysis.

The antimicrobial activity bioassays showed that from all tested strains, Aa3_DN216_4B10_1 was the only one with antibacterial activity, namely against MRSA. In fact, this strain was found to harbor the highest number of BGCs and ten-fold more NRPS/NRPS-like BGCs compared to the rest of the isolates (Figure 3 and Appendix A). Given the relatively large number of contigs and NRPS/NRPS-like BGCs, further inspection of the BGC regions showed that four out of ten were located at a contig edge. Even though this might indicate the presence of fragmented BGCs across multiple contigs [50], strain Aa3_DN216_4B10_1 showed the highest abundance of NRPS/NRPS-like BGCs among all tested ones. Additionally, the KnownClusterBlast tool of antiSMASH identified three BGCs (BGC 16, BGC 4, and BGC 3) with 100% similarity to known clusters in the Minimum Information about a Biosynthetic Gene (MIBiG) database related to the production of heterobactin A/heterobactin S2 (MIBiG:BGC0000371), rhizomide A (MIBiG:BGC0001758) and branched-chain fatty acids (BCFAs) (MIBiG:BGC0001534) (Appendix A). The structure of the aforementioned BGCs are displayed in Appendix A. Based on the antiSMASH analysis, the NRPS cluster predicted to code for heterobactin A/heterobactin S2 consisted of 11 genes responsible for core and additional biosynthesis and transport. It had a well-defined modular structure consisting of all key NRPS components such as adenylation, condensation, thiolation and certain tailoring domains. The NRPS-like BGC potentially encoding the lipopeptide rhizomide A was located at the edge of the contig and contained an adenylation and a thiolation domain. In the case of BGC 3, antiSMASH classified it as bacteriocin type with a predicted core structure for the production of bacteriocins next to the machinery for synthesizing BCFAs. Specifically, genes belonging to BGC 3 were 100% similar to the known *bkd* BGC in *Streptomyces filamentosus* (MIBiG:BGC0001534). Other BGCs homologous to those coding for known antimicrobials, such as chloramphenicol (MIBiG:BGC0000893), kirromycin (MIBiG:BGC0001070), and bacillomycin D (MIBiG:BGC0001090), were also detected in its genome but with lower similarity (< 20%) (Appendix A).

Comparative analysis of the 21 marine bacterial genomes in terms of their BGCs showed that three strains of the top five with the largest number of BGCs also had the highest activity against the tested cancer cell lines. According to the experimental results, Pf1_DN64_8G1, Aa3_DN213_3F7, and Aa3_DN64_1D3 induced cell death to all five cancer cell lines with average cytotoxicity of 67.8%, 48.7% and 43.3%, respectively. Pf1_DN64_8G1 and Aa3_DN64_1D3 harbored ten BGCs each, from which only three BGCs had homologs encoding known natural products (Appendix A). Two of these clusters were identified as beta-lactone BGCs and showed low similarity to those coding for pseudaminic acid (MIBiG:BGC0001747) (22%) and fengycin (MIBiG:BGC0001095) (13%), compounds known for their cytotoxic activity [51,52,53]. On the other hand, seven out of nine BGCs of Aa3_DN213_3F7 gave hits to the MIBiG repository (Appendix A). Similarly, four BGCs were related to clusters involved in the production of substances with reported anticancer activity: carotenoids (MIBiG:BGC0000636) (85% similarity) [54,55], ectoines (MIBiG:BGC0000853) (75% similarity) [56] and the siderophore desferrioxamine E (MIBiG:BGC0001478) (50% similarity) [57] (Appendix A).

## 3. Discussion

Sponge-associated microbes have already been acknowledged as important members of the “Natural Products Hall of Fame” for the production of substances with various bioactivities [35]. Nevertheless, the majority remains recalcitrant to in vitro cultivation hindering the flow in the marine drug biodiscovery pipeline [58]. Intensive research has focused on screening methods and subsequent elucidation of the chemical molecules. To shed more light on the bioactive potential of microbes, integration of genome mining with activity-driven screenings could uncover the true metabolite arsenal of the microbes of interest [41].

In the present study, we initially obtained crude extracts from 21 bacterial strains isolated from marine sponges and determined their antibacterial and cytotoxic activity via high-throughput screening. 16S rRNA gene- and whole genome-based phylogenetic analysis revealed that the tested strains belong mainly to four phyla, Proteobacteria, Actinobacteria, Bacteroidetes, and Firmicutes (Figure 1). Several of these taxa have been highlighted before for their antimicrobial activity. Actinobacteria and Proteobacteria are considered the most prolific sources of bioactive secondary metabolites with a wide spectrum of bioactivities among marine microorganisms [59,60,61,62,63]. However, only one strain, identified as *Rhodococcus erythropolis*, possessed anti-MRSA potential. To the best of our knowledge, no anti-MRSA activity has been previously detected in Rhodococci derived from marine sponges. Chelossi et al. [64] and Abdelmohsen et al. [65] retrieved *Rhodococcus* strains from *Petrosia ficiformis* and an unidentified marine sponge, respectively, with the ability to inhibit growth of *S. aureus* but not the methicillin-resistant strain. A possible explanation for the general absence of antimicrobial activity of the strains tested here may be the use of crude extracts as testing material and thus, the active principles being present in low concentrations. Moreover, eight of the *Pseudovibrio* strains screened here were selected because they were previously found to be resistant against several antibiotics, such as ampicillin, vancomycin, and tetracycline [44]. Yet, these antibiotic resistance phenotypes did not match with the antimicrobial activity screening results, which could be due to a lack of induction of gene expression under the tested conditions. Diverse fermentation conditions using various nutrients and treatments (e.g., co-culturing) may be of great value in activating silent genes in the search for bioactive secondary metabolites [41,60].

The high potential of marine sponge derivatives to inhibit tumor proliferation has led to increasing research efforts towards the discovery of new anticancer compounds [66,67,68]. To date, more than 10% of the screened marine sponges display cytotoxicity against human cancer cell lines [66]. Here, the effect of the crude extracts was more evident on HepG2 cells compared to other cancer cell lines (Figure 2). Among the tested strains, we could distinguish four extracts obtained from strains belonging to the genera *Pseudovibrio*, *Psychrobacter,* and *Brevibacterium* that showed the highest activity mainly against A2058, A549, and MiaPaca2 cancer cells. The inhibitory effect on A549 cells by *Pseudovibrio* has previously been reported, and the responsible bioactive natural products were indole alkaloids identified in cultures of *Pseudovibrio denitrificans* [69,70]. On the other hand, Choi et al. [71] reported *Brevibacterium*-derived compounds causing no induction of cell death on A549, AGC, MCF-7, and HepG2 carcinomas but only weak cytotoxicity against HL-60 cells.

To establish a link between the secondary metabolite BGC repertoire and the growth inhibiting effect against pathogenic bacteria and cancer cell lines, laboratory-based screening methods combined with genome mining were performed followed by GTM analysis. The top five strains in terms of BGC frequency and abundance consisted of three Proteobacteria and two Actinobacteria strains, confirming their strong bioactivity potential observed before [44,72,73,74]. Nevertheless, the abundance and diversity of BGCs were not reflected in the phenotype of the strains as only one displayed antibacterial activity under the tested conditions. Anti-MRSA activity shown by the *Rhodococcus* strain (Aa3_DN216_4B10_1) could be explained by the distinctively higher number of NPRS-encoding BGCs in its genome indicating a large specialized secondary metabolite arsenal [75,76,77]. Two NRPS BGCs had 100% homology with those coding for the siderophore heterobactin A/heterobactin S2 and the lipopeptide rhizomide A, respectively. Siderophores are small molecules that beyond iron acquisition were recently suggested to act as virulence factors and regulators of pathogenicity [78]. In addition, Schneider et al. [79] isolated a siderophore from the co-culture of two marine Proteobacteria, which displayed species-specific toxicity towards *S. aureus* without affecting MRSA. Likewise, in the case of rhizomide A, antibacterial activity has been observed against several clinically relevant strains, including *S. aureus* but not MRSA [80]. Another potential candidate responsible for the anti-MRSA activity of the *Rhodococcus* strain might be a putative peptide that contains branched-chain fatty acyl groups encoded by BGC 3 (Appendix A). Part of this BGC is 100% homologous to the *bkd* BGC in *Streptomyces filamentosus,* which encodes the branched-chain α-keto acid dehydrogenase (BKDH), a multi-subunit enzyme complex critical in the synthesis of BCFAs. These fatty acids could serve as precursors in the biosynthesis of antibiotics, such as daptomycin [81], which is currently one of the main treatment options for MRSA infections [82,83].

Interestingly, the experiments with human cancer cells showed that three out of the five strains with the highest cytotoxicity harbored the largest number of BGCs, suggesting a genotype-phenotype causality. Specifically, the most bioactive extracts belonged to *Pseudovibrio* strains (Pf1_DN64_8G1 and Aa3_DN64_1D3), which carried in their genomes two beta-lactone BGCs similar to those coding for natural products with known anticancer effect, namely pseudaminic acid and fengycin. Kokoulin et al. [51] were the first to report a capsular polysaccharide containing pseudaminic acid from another proteobacterium (*Psychrobacter marincola*), which significantly inhibited the growth of HL-60 cells. Fengycins are biosurfactants that have been studied in detail for their bioactivities and considered important inhibitors in cancer research [84]. Yet, the only marine-derived fengycin with anticancer potential is a fengycin isoform isolated from *Bacillus circulans* with a high efficacy against human colon carcinoma cells [85]. However, the similarity of the BGCs with those of known activity was relatively low, 22% and 13% for the pseudaminic acid and the fengycin BGC, respectively. This could be explained by the fact that the majority of BGCs detected in the two *Pseudovibrio* genomes did not share any similarity with characterized BGCs highlighting their unknown function in accordance with previous studies [44,73]. On the other hand, most BGCs of Aa3_DN213_3F7 had homologs with known bioactivity that confirms the extensive research focus on Actinobacteria in terms of their natural products [63]. The BGCs in the genome of strain Aa3_DN213_3F7 with the highest similarity were related to BGCs coding for carotenoids, ectoines, and the siderophore desferrioxamine E. Based on our experimental results, the highest cytotoxicity of Aa3_DN213_3F7 was observed against skin, lung, and hepatic carcinomas. Several studies have described the antiproliferative effect of marine carotenoids on different cancer cell lines, such as breast, intestinal, hepatic, and leukemic [54]. The carotenoid BGC was 85% similar to a BGC identified in *Brevibacterium linens* that encodes a novel lycopene cyclase, which catalyzes the biosynthesis of β-carotene from lycopene [86]. β-carotene is a successful carotenoid in the global market with many industrial applications, including prevention of cancer [55]. In the case of ectoines, Sheikhpour et al. [56] showed that ectoine and hydroxyectoine isolated from *Streptomyces* induced apoptosis in lung cancer cells. Another compound involved in the cytotoxicity of Aa3_DN213_3F7 could be related to desferrioxamine E, a synonym for nocardamine. It is a cyclic hydroxamic acid siderophore commercially available as antibiotic, anti-mycobacterial, iron-chelating, antioxidant, and anticancer compound. According to previous studies, nocardamine isolated from a novel marine actinobacterium exhibited only antitumor effect but no cytotoxicity against human breast cancer and malignant melanoma cell lines [57]. Nevertheless, cancer cell toxicity high-throughput screening against the Canvass library of natural products revealed the antiproliferative effect of nocardamine on various cancer cell lines, including pancreatic and ovarian [87].

## 4. Materials and Methods

### 4.1. Isolation of Strains and Growth Conditions

Twenty one bacterial isolates were obtained from six different sponge species (*Aplysina aerophoba*, *Petrosia ficiformis*, *Corticium candelabrum*, *Ircinia* sp., *Chondrilla nucula,* and *Acanthella acuta*), in previous studies [44,45].

Cryopreserved glycerol stocks of the strains were initially used as inoculum for regrowth on the original solid isolation media (Appendix A) at 20 °C. Single colonies were picked and cultured in 250 mL Erlenmeyer flasks containing 20 mL of the respective liquid media in duplicates. The flasks were incubated at 20 °C and shaken at 150 rpm in the dark for seven days. OD600 measurements were taken every 24 h in order to monitor the cell growth. After seven days of incubation and all cultures reaching the stationary phase, the content of the flask was stored at −20 °C until chemical extraction.

### 4.2. Strain Identification and Sanger Sequencing

The identity of the strains was confirmed by 16S rRNA gene Sanger sequencing. Single colonies were picked, stored in 100 μL nuclease-free water at −20 °C and served as template. The 16S rRNA gene amplicons were generated by PCR using primers 27F (5′-AGAGTTTGGATCMTGGCTCAG-3′) and 1492R (5′-CGGTTACCTTGTTACGACTT-3′) [88]. PCR reaction mixture and conditions were the same as described earlier [89]. PCR products were purified using the CleanPCR kit (CleanNA, Waddinxveen, The Netherlands) and quantified using a Nanodrop 2000c spectrophotometer (Thermo Fisher Scientific, Waltham, MA, USA). The purified PCR products were sequenced at Eurofins Genomics (Cologne, Germany) with primers 27F and 1492R. Quality control of the raw sequences was done with BioEdit 7.2.5 [90] by inspecting the chromatograms. In addition, consensus sequences of the near full length 16S rRNA gene were obtained by aligning the forward and reverse read with default settings. Taxonomic classification based on the near full length 16S rRNA gene sequences was performed using the blastn suite [91] against the NCBI nr/nt database (accessed on November 2020), considering as best hits the ones belonging to cultured representatives (Table 1 and Appendix A).

### 4.3. Chemical Extraction

The culture broths from the Erlenmeyer flasks were extracted with the use of Sepabeads^®^ SP207SS resin (Sorbent Technologies, Inc., Norcross, GA, USA). In particular, 20 mL of acetone was added to 10 mL of culture broth in EPA vials (Dispolab, Someren, The Netherlands). After 2 h of shaking at 220 rpm and room temperature (Kuhner ISF4-X Climo-Shaker, Kuhner Shaker S.A., Barcelona, Spain), the solvent was evaporated under a nitrogen stream overnight. Distilled water until final volume of 10 mL and 3 mL of a suspension of SP207ss resin dissolved in water were added to each sample. The vials were then placed in the incubator shaker at 220 rpm and room temperature for 2 h. A centrifugation step followed to settle the resin at 3500× *g* for 15 min (Speed Vac Plus SC210A, Savant Instruments, Inc., Holdbrook, NY, USA) and supernatants were discarded. Resin was washed by adding 16 mL of distilled water into each vial followed by shaking at 220 rpm and room temperature for 2 h. Samples were again centrifuged at 3500× *g* for 15 min and supernatants were discarded. To extract the adsorbed molecules from the resin, 10 mL of acetone was added to the resin for overnight mixing (Kuhner ISF4-X Climo-Shaker) at 20 °C. A last step of centrifugation (at 3500× *g* for 15 min, Speed Vac Plus SC210A, Savant Instruments, Inc., Holdbrook, NY, USA) was performed and the acetone extracts (organic phase) were transferred to glass tubes where 100% DMSO was added, and evaporated overnight under a nitrogen stream to 20% DMSO and a concentration of 2 × WBE (Whole Broth Equivalent). Moreover, 500 μL aliquots were prepared for each sample and stored at 4 °C overnight until bioactivity screening.

### 4.4. Bioactivity Screening Assays

#### 4.4.1. Antimicrobial Activity Test

The antimicrobial activity of the crude extracts was assessed against a panel of five different pathogenic microorganisms available from the Culture Collection of Fundación MEDINA (Granada, Spain). Antibacterial activity assays were performed against the Gram-negative *Escherichia coli* ATCC 25922 and *Klebsiella pneumoniae* ATCC 700603 and the Gram-positive methicillin-resistant *Staphylococcus aureus* MRSA MB5393. Standard drugs for the antibacterial bioactivity assays were aztreonam for *E. coli* ATCCC 25922, gentamicin sulfate for *K. pneumoniae* ATCC 700603, and vancomycin for *S. aureus* MRSA MB5393. To assess the antifungal activity, the fungi *Candida albicans* ATCC 64124 and *Aspergillus fumigatus* ATCC 46645 were used. Amphotericin B was used as positive control for the antifungal bioactivity assays. All bioassays were conducted according to previously described methodologies [92,93,94,95]. The final volume in the assay was 100 μL. No serial dilutions were used as the testing material was crude extract and not a pure compound. Incubation time was 18 h and temperature 37 °C. The assay was done in duplicate in different days. Absorbance (at 600 nm for bacterial strains and 612 nm for *C. albicans*) or fluorescence (excitation 570 nm, emission 615 nm for resazurin for *A. fumigatus*) were measured using an EnVision 2104 multi-label plate reader (PerkinElmer Inc., Waltham, MA, USA). Data analysis was conducted using the Genedata Screener^®^ software 7.0 (Genedata, Inc., Basel, Switzerland). Percentage of growth inhibition was calculated as reported by Martin et al. [93].

#### 4.4.2. Cell Viability Assay

The cytotoxic activity of the extracts was tested by MTT (3-(4,5-dimethylthiazol-2-yl)-2,5-diphenyltetrazolium bromide) assays against the following human cancer cell lines: lung carcinoma A549 (ATCC^®^ CCL-185™), melanoma A2058 (ATCC^®^ CRL-11147™), liver hepatocellular carcinoma HepG2 (ATCC^®^ HB-8065™), breast adenocarcinoma MCF7 (ATCC^®^ HTB-22™) and pancreas carcinoma MiaPaca-2 (ATCC^®^ CRL-1420™). All cell lines were sourced from the American Type Culture Collection (ATCC, Manassas, VA, USA). The cells were cultured in 96-well plates at a cell density of 10.000 cells/well and maintained at 37 °C, 90% humidity and 5% CO_2_. A total of 5 μL of each extract dispensed in 195 μL of medium were used as inoculum for the anticancer assay. The plates were then incubated for 72 h at 37 °C, 90% humidity and 5% CO_2_. Methyl methanesulfonate (MMS) 8 mM was used as positive control and 20% DMSO as negative control. A dose-response curve using doxorubicin (chemotherapeutic agent of natural origin) was also included as control, starting at 5 mM with an 8-point serial dilution (1/3). After 72 h treatment, the plates were washed with 100 μL of 1x PBS per well using a Multidrop™ Combi Reagent Dispenser (Thermo Fisher Scientific, Waltham, MA, USA). The MTT solution (tetrazolium dye, 0.5 mg/mL, final concentration 100 μL/well) was then added and the plates were incubated for approximately 3 h at 37 °C. Supernatants were discarded and 100 μL of 100% DMSO was added to each well to dissolve the formazan precipitates. Absorbance levels were measured at 570 nm using a Wallac 1420 Victor2™ Microplate Reader (WALLAC Oy PerkinElmer, Turku, Finland) and the resulting data was analyzed using the Genedata Screener^®^ software 7.0 (Genedata, Inc., Basel, Switzerland). The test was performed in triplicate.

### 4.5. Genomic DNA Extraction and Whole Genome Sequencing

Overnight liquid bacterial cultures were generated by picking up the same colony used as template for Sanger sequencing and using this to inoculate the respective media in 50 mL tubes at 20 °C (Appendix A). Genomic DNA was extracted from 1 mL of overnight cultures using the MasterPure™ Gram-Positive DNA Purification Kit (Lucigen, Epicentre). Concentration and purity of the extracted DNA were measured using a Nanodrop 2000c spectrophotometer (Thermo Fisher Scientific, Waltham, MA, USA) and a Qubit dsDNA BR Assay kit (Invitrogen, Thermo Fisher Scientific, Waltham, MA, USA) used with a DS-11 FX Fluorometer (DeNovix, Inc., Wilmington, DE, USA).

Whole genome sequencing of eight strains (Aa3_DN64_1D3, Aa3_Str.68_7G12, Acac_Ps_AB113, Cn_Ps_AB111, Irc_Ps_AB108, Pf1_DN206_4B7, Pf1_Ps_8H04_1, Pf1_Ps_8H06) was performed previously using the Illumina MiSeq platform (paired end, 2 × 300 bp) at GATC Biotech (Konstanz, Germany; now part of Eurofins Genomics Germany GmbH) in a previous study [44]. The remaining 13 strains were sequenced in the framework of this study with Illumina HiSeq (paired end, 2 × 150 bp reads) at Novogene Europe (Cambridge, UK).

### 4.6. Bioinformatic Analysis

#### 4.6.1. Quality Control and Genome Assembly

The quality of the Illumina HiSeq reads was investigated with FASTQC 0.11.9 [96]. Adapter removal and read quality filtering was performed with Trimmomatic 0.39 [97] with the following settings: ILLUMINACLIP:/Adapters.fa:2:30:10, HEADCROP:10, MINLEN:40, SLIDINGWINDOW:4:15. Illumina HiSeq reads were de novo assembled with SPAdes 3.14.0 [98], and contigs less than 1000 bp long were filtered out using reformat.sh from BBTools Suite 38.84 [99]. All draft assemblies were checked for the presence of contigs assigned to sequencing artifacts, which were subsequently discarded. Mapping of the quality-filtered reads to the assembled contigs with Bowtie 2.4.1 [100] using default setting followed. The resulting sequence alignment map (SAM) file was converted into a binary alignment map (BAM) file, which was sorted and then indexed with SAMtools 1.10 [101]. Per base coverage of the draft assemblies was calculated with the “genomecov” command of BedTools 2.29.1 [102] using the sorted, indexed BAM file as input. Quality of all draft assemblies was assessed with QUAST 5.0.2 [103] and completeness and contamination were evaluated with CheckM 1.1.2 [104] with the default set of marker genes. All relevant information regarding the Illumina MiSeq reads processing and genome assembly are described in Versluis et al. [44]. Raw read sequences and draft genome assemblies generated in this study have been deposited in the European Nucleotide Archive (ENA) under the study accession number PRJEB41620.

#### 4.6.2. Phylogenetic Analysis and Genome Mining

The GTDB-Tool Kit 1.1.0 (GTDB-Tk) [105] was used to taxonomically classify all strains based on the presence of single-copy marker genes in their draft assemblies and the placement of their genomes in the Genome Taxonomy Database (GTDB) reference tree [48,106]. The resulting concatenated alignment of the translated amino acid sequences of 120 bacterial marker genes identified in the draft assemblies was used to generate a maximum likelihood tree using FastTree 2.1.11 [107] with default parameters. Tree visualization was done with Interactive Tree of Life (iTOL) version 3 [108]. For annotation of secondary metabolite Biosynthetic Gene Clusters (BGCs) in the draft assemblies, the online server antiSMASH 5.0 [49] was employed with “relaxed” detection strictness and all extra features activated.

### 4.7. Data Visualization and Availability

Plots were created in R Studio with R version 3.5.0 [109] using the R package ggplot2 version 3.3.2 [110]. All codes and data used for the genomic analysis and data visualization can be found in https://github.com/mibwurrepo/Gavriilidou_et_al_2021_Bioactivity_Screening.git.

## 5. Conclusions

The bioactivity screening and subsequent phylogenetic analysis of the sponge-associated bacteria revealed that the most active extracts both in terms of antibacterial and anticancer activity were affiliated with the phyla Actinobacteria and Proteobacteria, supporting these two taxonomic groups as prominent sources of bioactive substances. Yet, the antiproliferative impact on human pathogens was less pronounced than that on cancer cells. This is in contrast with the abundance of BGCs detected in the genomes and points out the need of implementing diverse cultivation regimes to trigger the expression of the appropriate genes linked to the production of bioactive molecules. On the other hand, the most cytotoxic strains were among the ones with the highest number of BGCs. GTM analysis revealed several BGCs related to compounds potentially responsible for the induction of cell death of the respective cancer cell lines, facilitating the distinction of favorable candidates. Altogether, these findings highlight the importance of integrating phenotypic assays with genome mining in order to provide insights on the most promising leads for further investigation, such as isolation and identification of the active principles.

## Figures and Tables

**Figure 1 marinedrugs-19-00075-f001:**
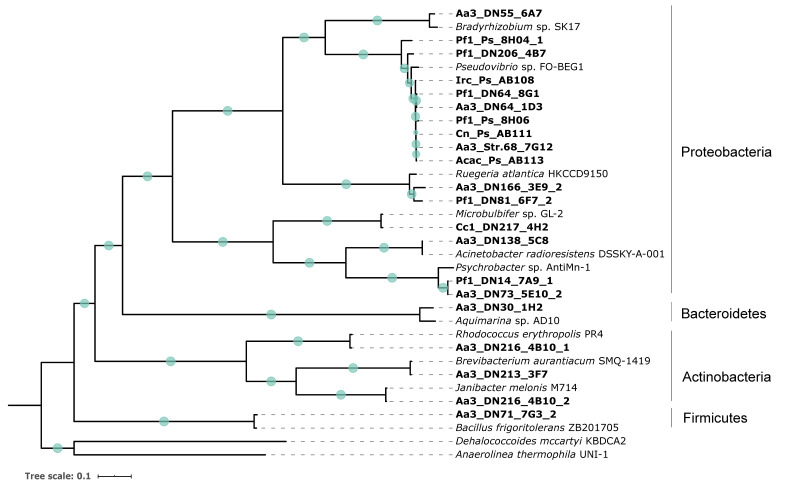
Maximum likelihood tree inferred from a concatenated alignment of 120 conserved amino-acid sequences of bacterial marker genes. Green circles on the branches display bootstrap values (>70%). Strains included in the present study are indicated in bold. *Dehalococcoides mccartyi* KBDCA2 and *Anaerolinea thermophila* UNI-1, both members of the Chloroflexi, were used as outgroup. The scale bar represents the number of estimated substitutions per site.

**Figure 2 marinedrugs-19-00075-f002:**
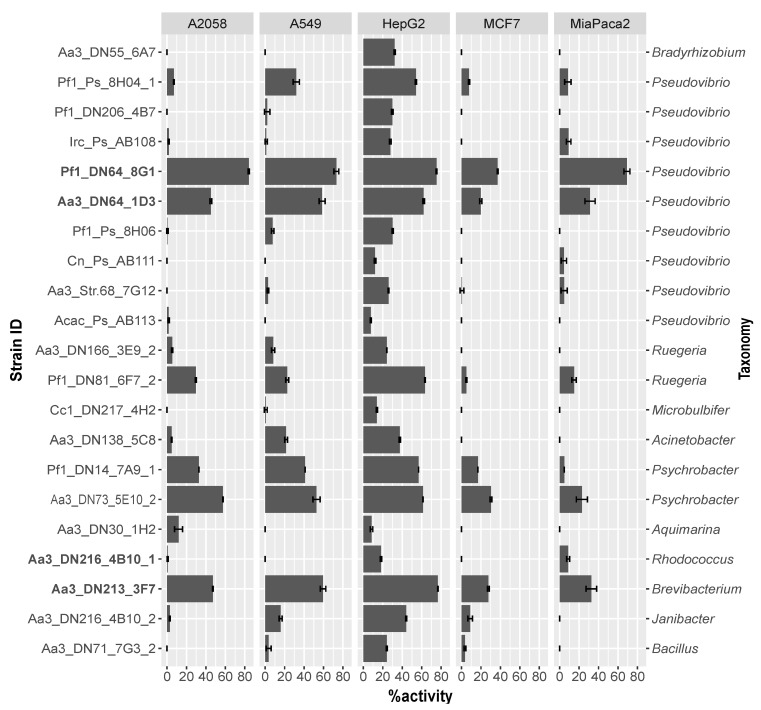
MTT (3-(4,5-dimethylthiazol-2-yl)-2,5-diphenyltetrazolium bromide) assay results. Percentage of activity or cell death of five human cancer cell lines (A2058: melanoma, A549: lung carcinoma, HepG2: hepatocyte carcinoma, MCF7: breast adenocarcinoma and MiaPaca2: pancreas carcinoma) after 72 h of incubation with crude extracts of the tested strains in triplicate. Error bars represent standard errors. Strains are ordered from top to bottom in accordance with the phylogenetic tree in Figure 1 and in bold are the ones selected for gene-trait matching.

**Figure 3 marinedrugs-19-00075-f003:**
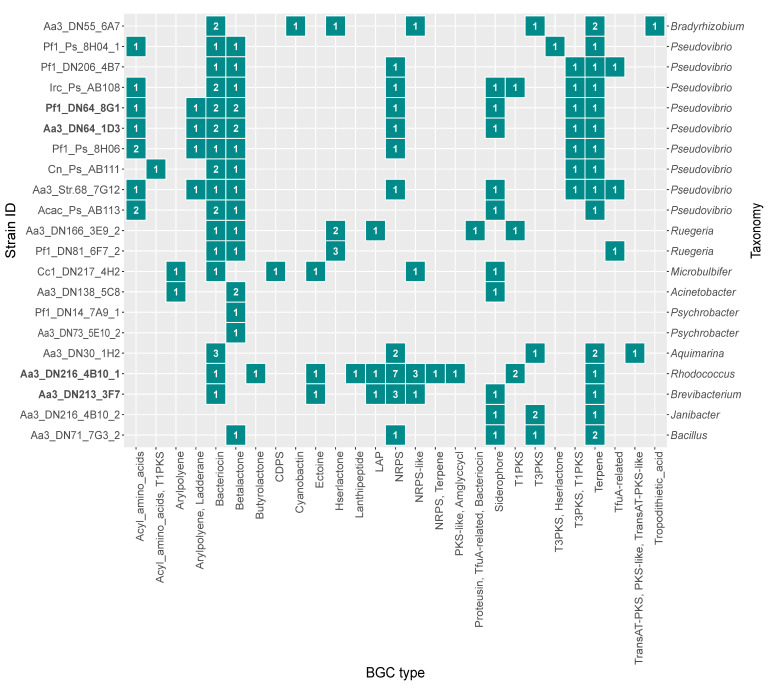
Absolute abundance of secondary metabolite biosynthetic gene clusters (BGCs) predicted in the genomes of the strains screened for their bioactivity in the present study. Strains selected for gene-trait matching are marked in bold.

**Table 1 marinedrugs-19-00075-t001:** Information on taxonomy of sponge-associated isolates according to BLAST searches of their partial 16S rRNA gene sequences against nr/nt NCBI database and genome characteristics. All details on classification of strains and genome assembly statistics are provided in the Appendix A. Order of strains is in accordance with the phylogenetic tree in Figure 1. Genomes of marked strains were generated in a previous study [44].

Strain ID	Isolation Source	Best BLAST hit	ID%	Genome Size (Mbp)	GC Content (%)	Total Gene Count
Aa3_DN55_6A7	*Aplysina aerophoba*	*Bradyrhizobium* sp.	100	7.2	64.6	6655
Pf1_Ps_8H04_1 ^1^	*Petrosia ficiformis*	*Pseudovibrio* sp.	99.9	5.7	48.2	5174
Pf1_DN206_4B7 ^1^	*Petrosia ficiformis*	*Pseudovibrio* sp.	99.85	5.1	52.8	4692
Irc_Ps_AB108 ^1^	*Ircinia* sp.	*Pseudovibrio* sp.	100	5.9	44.6	5369
Pf1_DN64_8G1	*Petrosia ficiformis*	*Pseudovibrio* sp.	99.93	5.8	51.4	5275
Aa3_DN64_1D3 ^1^	*Aplysina aerophoba*	*Pseudovibrio* sp.	99.93	5.8	50.3	5239
Pf1_Ps_8H06 ^1^	*Petrosia ficiformis*	*Pseudovibrio* sp.	100	6.1	49.7	5554
Cn_Ps_AB111 ^1^	*Chondrilla nucula*	*Pseudovibrio* sp.	99.89	5.9	49.8	5423
Aa3_Str.68_7G12 ^1^	*Aplysina aerophoba*	*Pseudovibrio* sp.	100	5.9	51.0	5288
Acac_Ps_AB113 ^1^	*Acanthella acuta*	*Pseudovibrio* sp.	99.93	5.4	51.0	4886
Aa3_DN166_3E9_2	*Aplysina aerophoba*	*Ruegeria* sp.	99.93	4.6	56.2	4558
Pf1_DN81_6F7_2	*Petrosia ficiformis*	*Ruegeria atlantica*	100	4.5	57.9	4356
Cc1_DN217_4H2	*Corticium candelabrum*	*Microbulbifer echini*	99.85	4.7	49.8	4229
Aa3_DN138_5C8	*Aplysina aerophoba*	*Acinetobacter radioresistens*	100	3.3	41.4	3085
Pf1_DN14_7A9_1	*Petrosia ficiformis*	*Psychrobacter celer*	100	2.9	46.8	2434
Aa3_DN73_5E10	*Aplysina aerophoba*	*Psychrobacter celer*	100	2.6	47.0	2194
Aa3_DN30_1H2	*Aplysina aerophoba*	*Aquimarina macrocephali*	100	5.4	32.9	4684
Aa3_DN216_4B10_1	*Aplysina aerophoba*	*Rhodococcus erythropolis*	100	7.1	62.5	6770
Aa3_DN213_3F7	*Aplysina aerophoba*	*Brevibacterium aurantiacum*	100	4.2	63.0	3850
Aa3_DN216_4B10_2	*Aplysina aerophoba*	*Janibacter melonis*	99.93	3.4	72.9	3220
Aa3_DN71_7G3_2	*Aplysina aerophoba*	*Bacillus frigoritolerans*	100	4.9	40.5	4807

^1^ Modified from [44].

## Data Availability

The data presented in this study are available in the European Nucleotide Archive (ENA) under the study accession number PRJEB41620. All codes and scripts used for data analysis can be found here: https://github.com/mibwurrepo/Gavriilidou_et_al_2021_Bioactivity_Screening.git.

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
