# Peer review of "Bioactivity Screening and Gene-Trait Matching across Marine Sponge-Associated Bacteria"

_marinedrugs, 2021, doi:10.3390/md19020075_

Round 1

Reviewer 1 Report

The work submitted by Gavriilidou and colleagues "Bioactivity Screening and Gene-Trait Matching Across Marine Sponge-Associated Bacteria, combined experimental and computational approaches to screen sponge associated bacteria with potential bioactivities. Although the authors have performed experiments related to bioactivities and also sequenced the genomes of 21 bacterial strains, the data analysis especially related to genomic and BGCs is very simplistic, limited and not satisfactory. Therefore, the authors should address this major issue before the manuscript could be considered for publication.

My major concerns are regarding the BGC profiling in section 2.5. This is just a plain description of what anyone interested could see in an output of the antiSMASH pipeline.

Based on this reviewer personal experience, such description is sometimes simplistic, confusing or indicates obvious observations. The authors could have (and should have) analyzed the BGCs in details. Authors can have a look at the following papers on BGC analyses (https://doi.org/10.1186/s12864-020-6468-5; https://doi.org/10.3390/genes11101166; https://doi.org/10.1016/j.gene.2020.144379).

The authors should check if there are any trends in the distribution of BGCs across all 4 clades.

Line 169-171: The authors mention that antiMSASH predicted 3 BGCs with 100% similarity to know clusters from MIBiG database. However, no attempt was made to compare the architecture of such BGCs and the modular structure of the core enzymes.

All the genomes mentioned in the Table S1 are at the contig level, and in case the contigs are short, antiSMASH can detect a similar gene cluster on multiple contigs. Therefore, this could affect the total number of BGCs predicted by antiSMASH. For example:

Line 165-168: It is stated that strain Aa3_DN216_4B10_1 has highest number of BGCs including 10-fold more NRPS BGCs. However, the genome of this strain has also the second largest number of 79 contigs. Therefore, it is not clear if similar BGC is present in multiple contigs and reported two or more times. This information can be easily extracted from the output of antiSMASH and should be addressed in the manuscript.

Several BGCs (Table S3) are orphan clusters for which no similarity with known BGCs was found. The authors should have explored these orphan clusters and discussed them in the text.

Minor comments:

Table 1: Provide the best blast hits for each strain and the corresponding e-value and % identity scores.

Insert ‘comma’ after the first digit in the gene count column (e.g. change 6655 to 6,655)

Figure 1: Add at least phylum level annotation for 4 main clades. It will be easier for the readers to comprehend.

It is also not clear if the any of the genomes reported by the authors are already available in the public databases.

Author Response

Thank you for your comments, please see the attachment.

Reviewer 2 Report

This manuscript reports the isolation of marine bacteria from marine sponge, as well as antimicrobial and cytotoxic activities of the crude extracts of the isolated bacteria. This work obtained twenty-one sponge-associated bacteria, which were screened for antimicrobial and cytotoxic activities. Bacterial genomes were also mined for secondary metabolite biosynthetic gene clusters (BGCs). Phylogenetic analysis assigned the strains to four major phyla in the sponge microbiome including Proteobacteria, Actinobacteria, Bacteroidetes and Firmicutes. This work found one extract of the bacterium exhibiting anti-MRSA activity, and more than 70% of the total 18 extracts showing a moderate to high cytotoxicity. This manuscript is useful for readers in the field, and it is recommended for publication after minor revision. In order to improve this manuscript, please consider the comments and suggestions which are listed below.

  1. Recent reports on marine bacteria as important sources of bioactive compounds of pharmaceutical interests should be mentioned in the introduction. This would underscore the importance of the microorganisms investigated in this work. Please see: Marine Actinobacteria: Screening for Predation Leads to the Discovery of Potential New Drugs against Multidrug-Resistant Bacteria. Antibiotics 2020, 9, 91.; New marine-derived indolymethyl pyrazinoquinazoline alkaloids with promising antimicrobial profiles, RSC Adv., 2020, 10, 31187-31204.; From Ocean to Medicine: Pharmaceutical Applications of Metabolites from Marine Bacteria. Antibiotics 2020, 9, 455.; An antibiotic agent pyrrolo[1,2-a]pyrazine-1,4-dione,hexahydro isolated from a marine bacteria Bacillus tequilensis MSI45 effectively controls multi-drug resistant Staphylococcus aureus, RSC Adv., 2018,8, 17837-17846.
  2. “4.4.1. Antimicrobial Activity Test”; please provide names of standard drugs for this assay, both antibacterial and antifungal activities.

Reviewer 3 Report

Gavriilidou et al. sequenced the genomes of 21 bacterial symbionts of marine sponges and analyzed the number und nature of their biosynthetic gene clusters. The authors extracted metabolites from culture broth of the isolates and tested them for antibacterial, antifungal and cytotoxic activity. Lastly, they attempted to rationalize some of the experimental findings by gene-trait matching.

Overall, this manuscript is very well written and easy to follow. The methods are described in sufficient detail with a few exceptions (see comments below). The experiments were conducted carefully and analyzed appropriately. Speculative interpretations are clearly marked as such. I have no major concerns about this manuscript. The only (major) question/suggestion I have, is whether the authors also used chromatographic methods and mass spectrometry to assess the content of the tested extracts. At least for the strain with antimicrobial activity, it would be interesting to see if any of the putative products of the BGCs listed were detected.

Minor concerns:

Line 89 and corresponding methods section. Please be more specific of when and how these bacteria were isolated. In the current version I cannot tell if the isolation was part of this study or part of a previous study. If the latter is the case, please use appropriate references in these sections and Table 1.

Line 207: “A possible…” are the authors referring to their own work or to the work cited in the previous sentence? Please change to make it clearer.

Line 214: Maybe the argument of the silent clusters should be mentioned in the same sentence as the potential low concentrations.

Line 220: “selective activity” of what against what? I think it’s very odd that basically all extracts show some toxicity on this cell line. It would be interesting to see how the extracts perform against other liver cell lines.

Line 330: please provide at least the scale of the assay, incubation time and temperature etc. Please provide the number of biological replicates.

Round 2

Reviewer 1 Report

I have no further comments.

Author Response

We would like to thank Reviewer 1 for the constructive feedback and the important suggestions in improving our manuscript.